# Impaired extinction of cocaine seeking in HIV-infected mice is accompanied by peripheral and central immune dysregulation
Lauren A. Buck[1], Qiaowei Xie[1,2], Michelle Willis[1], Christine M. Side[1], Laura L. Giacometti[1], Peter J. Gaskill[1], Kyewon Park[3], Farida Shaheen[3], Lili Guo[4], Santhi Gorantla[4] & Jacqueline M. Barker [1] ✉

Substance use disorders (SUDs) are highly comorbid with HIV infection, necessitating an understanding of the interactive effects of drug exposure and HIV. The relationship between HIV infection and cocaine use disorder is likely bidirectional, with cocaine use directly impacting immune function while HIV infection alters addiction-related behavior. To better characterize the neurobehavioral and immune consequences of HIV infection and cocaine exposure, this study utilizes a humanized mouse model to investigate the outcomes of HIV-1 infection on cocaine-related behaviors in a conditioned place preference (CPP) model, and the interactive effects of cocaine and HIV infection on peripheral and central nervous system inflammation. HIV infection selectively impairs cocaine CPP extinction without effecting reinstatement or cocaine seeking under conflict. Behavioral alterations are accompanied by immune changes in HIV infected mice, including increased prefrontal cortex astrocyte immunoreactivity and brain-region specific effects on microglia number and reactivity. Peripheral immune system changes are observed in human cytokines, including HIV-induced reductions in human TNFα, and cocaine and HIV interactions on GM-CSF levels. Together these data provide new insights into the unique neurobehavioral outcomes of HIV infection and cocaine exposure and how they interact to effect immune responses.

An estimated 1.2 million people in the United States currently live with HIV infection[1]. Substance use disorder (SUD) is highly co-occurring with HIV infection. Cocaine use rates among people living with HIV (PLWH) are consistently considerably higher than those observed in the general population[2–9]. Both chronic exposure to cocaine and progressive HIV infection can independently alter behavior and immune system function and may interact to create a unique neurobehavioral niche. Cocaine use disorder is characterized by difficulty in terminating drug use, chronic relapse even after periods of abstinence, and persistent drug use despite adverse consequences. A greater understanding of the immune and behavioral consequences of HIV infection can enable development of targeted therapeutics to support cessation of cocaine use.

The relationship between drug exposure and HIV infection appears to be bidirectional. For example, cocaine has been demonstrated to enhance HIV replication and promote the development of HIV-associated neuro-cognitive disorders (HAND)[10–12], to elevate levels of proinflammatory cytokines in humanized mouse models prior to HIV-1 infection, and to accelerate infection[13]. In addition to cocaine effects on HIV outcomes, cocaine-related behaviors are dysregulated in HIV rodent models. A majority of the work has employed transgenic models to investigate outcomes in a cocaine conditioned place preference (CPP) paradigm which assesses the development and expression of drug-context associations. In HIV-1 Tat transgenic (Tg) mice, Tat expression potentiates cocaine CPP[14–16]. Inducing Tat expression is also sufficient to drive the reinstatement of cocaine CPP (refs. 14,15) following extinction, suggesting that HIV-1

[1]Department of Pharmacology and Physiology, Drexel University College of Medicine, Philadelphia, PA, USA. [2]Graduate Program in Pharmacology and Physiology, Drexel University College of Medicine, Philadelphia, PA, USA. [3]Center for AIDS Research, University of Pennsylvania, Philadelphia, PA, USA. [4]Medical Center, University of Nebraska, Omaha, NE, USA. ✉e-mail: jmb893@drexel.edu

proteins can act to alter cocaine-related behavior, potentially through inflammatory effects or effects on the dopamine transporter[17]. Notably, HIV-1 Tg rats show blunted cocaine preference and a failure to escalate self-administration[18], suggesting that there may be species or expression pattern specific differences in the effects of these proteins.

One challenge in preclinical studies of the outcomes of HIV infection is limitations in modeling progressive HIV infection[19]. There have been advances in rodent models of HIV, including the development of the EcoHIV model[20], which enables infection of wild type mice, and humanized models that can be successfully infected with HIV-1[21–25]. To investigate the effects of progressive HIV-1 infection on cocaine-related behavior and how interactions between HIV-1 and cocaine exposure impact peripheral and central inflammation, the current study utilized the CD34-NSG humanized mouse model, in which bone marrow engraftment enables humanization of the mouse immune system which can support productive HIV-1 infection. These experiments assessed cocaine-related behaviors in the CPP task, including preference, extinction, and reinstatement, and the expression of preference under conflict. The findings demonstrate that HIV-1-infected humanized mice form a preference for the cocaine-paired environment of similar magnitude to non-infected controls. However, HIV-infected mice exhibited greater preferences across extinction, indicating resistance to extinction of cocaine CPP. The results further demonstrate HIV-1 and cocaine effects on peripheral cytokine/chemokine levels, and further, microglial and astrocyte changes in key neuroanatomical substrates of reward-related memory. Together, these findings characterize HIV effects on cocaine-related behaviors and identify independent and interactive effects of HIV-1 infection and cocaine exposure on peripheral and central immune outcomes.

## Results

### HIV-1 infection of humanized mice

Adult male and female NOD/Scid IL2Rg-/- (NSG) mice were matched by engraftment status to undergo either sham or HIV inoculation and cocaine/saline exposure. There were no differences in initial human monocyte or lymphocyte composition between groups of mice, as confirmed by assessing expression of CD3 [HIV groups: F $(1,61) = 2.216$, $p = 0.1417$; cocaine exposure: F $(1,61) = 1.381$, $p = 0.2444$; Fig. 1b], CD14 [HIV groups: F $(1,61) = 0.7720$; $p = 0.3831$; cocaine exposure: F $(1,61) = 2.558$, $p = 0.1149$], and CD19 [HIV groups: F $(1,61) = 1.910$, $p = 0.1720$; cocaine exposure: F $(1,61) = 1.346$, $p = 0.2505$].

To determine HIV viral load, HIV-1 RNA was measured using Q-RT-PCR. An unpaired t test of RNA copies was performed to compare viral load in HIV and sham inoculated mice. There was a significant difference in mean HIV viral load between HIV inoculated mice and sham inoculated mice [$t(35) = 2.777$, $p = 0.0088$; Fig. 1c]. Viral load was assessed on alternating weeks following inoculation (Fig. 1d).

Mouse weights were collected three times per week across the study. In male cohorts, three-way ANOVA demonstrated no effects of HIV [$F(1,20) = 0.1002$, $p = 0.7549$; $\omega^2 = 0.4106$], cocaine [$F(1,20) = 0.1043$, $p = 0.7501$; $\omega^2 = 0.4273$], or time [$F(5.617,112.3) = 2.136$, $p = 0.0587$; $\omega^2 = 1.065$]; Geisser-Greenhouse corrected; Fig. 1e]. There were no effects of HIV [$F(1,27) = 0.8059$, $p = 0.3773$; $\omega^2 = 2.251$] or cocaine [$F(1,27) = 1.847$, $p = 0.1853$; $\omega^2 = 5.161$] on female mouse weights, however there was a main effect of time [three-way ANOVA; $F(6.952,187.7) = 2.776$, $p = 0.0092$; $\omega^2 = 1.407$; Greenhouse-Geisser corrected; Fig. 1f]. Post hoc comparisons revealed that weights were greater than day 0 on days 2, 29, 32, 36, 39, and 47 ($p$'s $< 0.05$).

### HIV infection does not alter cocaine preference in mice

In order to assess preference for the cocaine-paired chamber, mice were trained in alternating sessions to associate distinct environments with cocaine or saline. Across training, mice exhibited higher locomotion during cocaine sessions than saline sessions [two-way ANOVA $F(1,25) = 94.23$, $p < 0.0001$; $\omega^2 = 47.51$; Fig. 2a]. There was no effect of HIV infection on locomotion [two-way ANOVA; $F(1,25) = 2.590$, $p = 0.1201$; $\omega^2 = 3.443$] and no interaction [$F(1,25) = 0.006569$, $p = 0.9360$; $\omega^2 = 0.00312$]. Following training,

mice were tested for a cocaine preference in a 20-min session in which they were allowed to freely explore the CPP apparatus. CPP score was determined by comparing time spent in the cocaine-paired and unpaired chambers (time in paired – unpaired). Compared to baseline scores, all mice demonstrated a preference for the cocaine-paired chamber [two-way ANOVA $F(1,25) = 20.82$, $p = 0.0001$; $\omega^2 = 18.12$; Fig. 2b]. HIV infection did not impact the formation of a preference for the cocaine-paired chamber [two-way ANOVA; $F(1,25) = 0.06663$, $p = 0.4220$; $\omega^2 = 1.432$]. An unpaired t test was used to compare total locomotion in all three chambers on the CPP test day for cocaine-exposed mice. There was not a significant difference in total locomotion between sham and HIV-infected mice during the CPP test session [$t(25) = 1.462$, $p = 0.1563$, 95% C.I. = −154.9 to 912.3; Fig. 2c]. Together these data demonstrate that HIV infection did not impact acquisition or expression of cocaine conditioned place preference.

### HIV infection promotes persistent cocaine seeking across extinction and reinstatement

To determine the effect of HIV infection on extinction and stress-induced reinstatement of cocaine seeking mice were allowed three sessions without cocaine followed by a session in which mice were administered yohimbine prior to testing (Fig. 3a). HIV infection inhibited extinction such that HIV-infected mice continued to spend more time in the cocaine-paired chamber than sham-infected counter parts [two-way ANOVA; F $(1,24) = 4.913$, $p = 0.0364$; $\omega^2 = 7.59$; Fig. 3b]. A main effect of time was observed across sessions [F $(2.723,65.34) = 5.225$, $p = 0.0036$; Greenhouse-Geisser corrected; $\omega^2 = 9.343$].

Time spent in the cocaine-paired chamber during the reinstatement test did not differ significantly between sham and HIV-infected mice [two-way ANOVA; F $(1,24) = 4.170$, $p = 0.0523$; $\omega^2 = 10.67$; Fig. 3c]. No effect of time [F $(2.178,52.28) = 1.146$, $p = 0.3291$; $\omega^2 = 1.242$; Greenhouse-Geisser corrected], or interaction between HIV and time was observed [F $(3,72) = 0.3046$, $p = 0.2819$; $\omega^2 = 0.3301$].

### Sham and HIV-infected mice reduce cocaine seeking under conflict

To assess whether HIV infection impacted cocaine seeking despite adverse consequences, time spent in the reward paired chamber before and after a footshock were compared (Fig. 4a). A main effect of time was observed such that all cocaine-exposed mice reduced time in the cocaine-paired chamber following footshock [two-way ANOVA; F $(1,9) = 16.49$, $p = 0.0028$, $\omega^2 = 26.15$; Fig. 4b]. There was no effect of HIV [F $(1,9) = 1.945$, $p = 0.1966$, $\omega^2 = 10.56$] and no interaction [F $(1,9) = 0.4919$, $p = 0.5008$, $\omega^2 = 0.780$]. Latency to enter the cocaine-paired chamber was not altered after footshock [two-way ANOVA; F $(1,8) = 1.470$, $p = 0.2600$, $\omega^2 = 6.703$; data from one mouse excluded as an outlier; this did not impact conclusions]. There was no effect of HIV [F $(1,8) = 1.645$, $p = 0.2355$, $\omega^2 = 8.015$] and no interaction [F $(1,8) = 1.470$, $p = 0.2661$, $\omega^2 = 6.52$] observed (Fig. 4c). An unpaired t test was used to compare total locomotion in all three chambers on the CPP test day for cocaine-exposed mice. There was not a significant difference in total locomotion between sham and HIV-infected mice during the CPP test session [$t(9) = 0.9337$, $p = 0.3748$, 95% C.I. = −175.6 to 422.5; Fig. 4d].

### GFAP immunoreactivity is increased in the PFC of HIV-infected mice

To determine if HIV infection or cocaine exposure altered GFAP immunoreactivity, brain tissue was analyzed for the percent area of GFAP staining in reward substrates (Fig. 5a). A two-way ANOVA revealed an increased percent area of GFAP staining in HIV-infected mice in the prelimbic [main effect of HIV: F $(1,28) = 5.619$, $p = 0.0249$; $\omega^2 = 15.73$; Fig. 5b] and infralimbic [main effect of HIV: F $(1,28) = 4.447$, $p = 0.0440$; $\omega^2 = 13.32$; Fig. 5c] PFC, but no effect of cocaine [PrL: F $(1,28) = 1.733$, $p = 0.1998$, $\omega^2 = 4.851$; IL: F $(1,28) = 0.5684$, $p = 0.4572$; $\omega^2 = 1.702$] and no interaction [PrL: F $(1,28)$ 0.3613, $p = 0.5526$, $\omega^2 = 1.012$; IL: F $(1, 28) = 0.3745$, $p = 0.5455$; $\omega^2 = 1.122$] were observed.

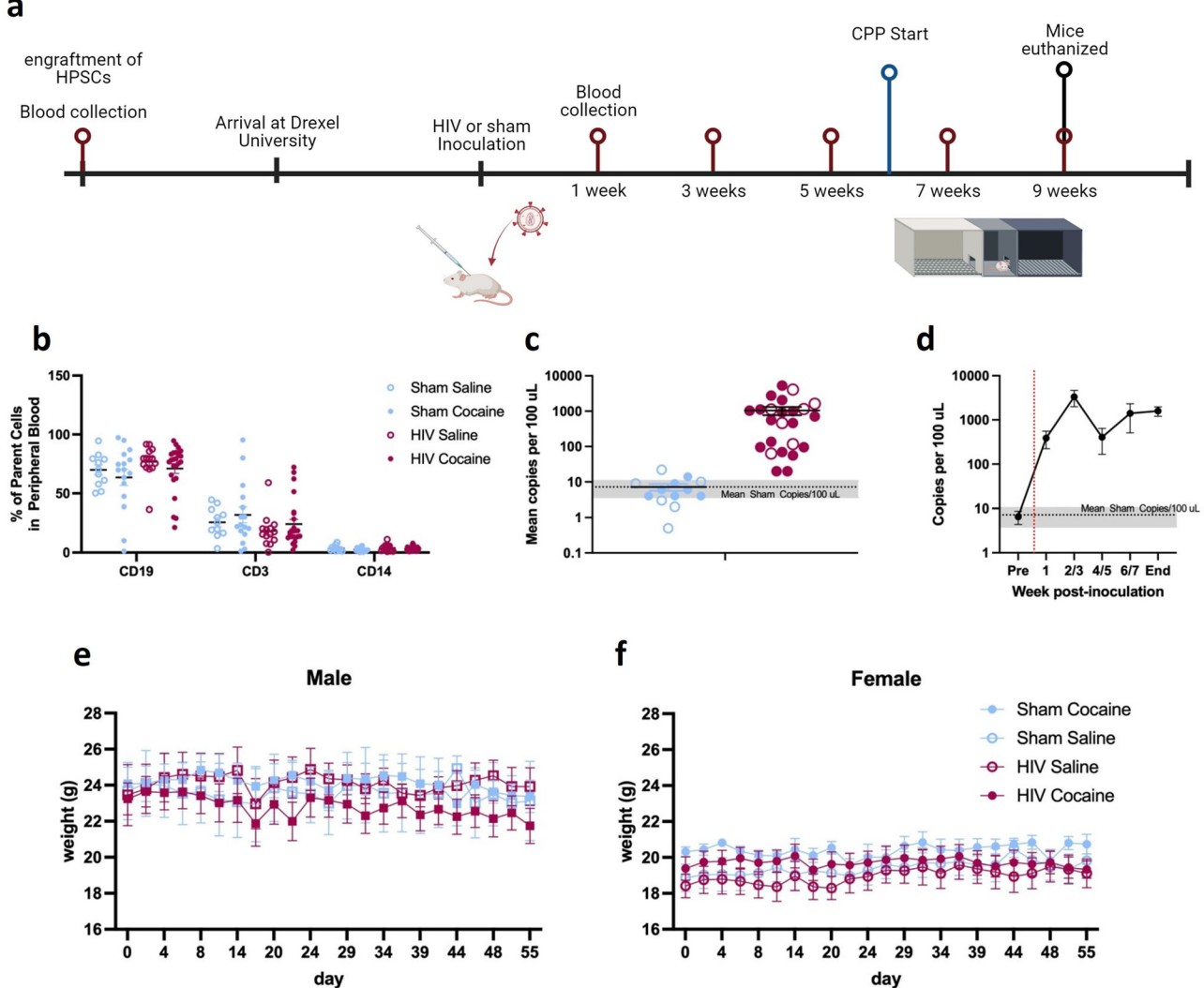

**Fig. 1 | Humanized mouse model of HIV-1 infection.** Mice were matched based on engraftment status to undergo HIV-1 or sham infection and subsequent cocaine or saline treatment. **a** Timeline of experiments (created with Biorender.com). **b** Initial human monocyte and lymphocyte composition was assessed using CD19, CD3, and CD14 among mice that were later assigned as sham-saline (n = 11), sham-cocaine (n = 16), HIV-saline (n = 14), or HIV-cocaine (n = 24). **c** The overall mean number of copies of HIV-1 RNA per 100 μl of plasma sample was elevated in HIV-1 inoculated mice (n = 24) compared to Sham (n = 13) mice. Gray shaded area indicates mean results of Sham inoculated animals and 95% confidence interval. **d** Number of copies of HIV-1 RNA per 100 μl of plasma in HIV-1 inoculated mice across the experimental timeline. Red dashed line indicates inoculation time point. Gray shaded area indicates mean results from Sham animals and 95% confidence interval. No effect of HIV-1 infection or cocaine treatment were observed on weight (grams) of male mice (n = 24; **e**) or female (n = 31; **f**) mice throughout the duration of the experiment. *$p < 0.05$. Points represent mean ± SEM. Open and closed symbols represent saline and cocaine exposed mice, respectively.

No effects of cocaine or HIV infection on GFAP immunoreactivity were observed in the hippocampus [interaction: $F_{(1,36)} = 3.124$, $p = 0.0856$, $\omega^2 = 7.692$; main effect of cocaine: $F_{(1,36)} = 0.5794$, $p = 0.4515$, $\omega^2 = 1.427$; main effect of HIV: $F_{(1,36)} = 0.8573$, $p = 0.3607$, $\omega^2 = 2.111$; Fig. 5d], the nucleus accumbens [interaction: $F_{(1,20)} = 0.7723$, $p = 0.3900$, $\omega^2 = 3.604$; main effect of cocaine: $F_{(1,20)} = 0.5910$, $p = 0.4510$, $\omega^2 = 2.758$; main effect of HIV: $F_{(1,20)} = 0.04610$, $p = 0.8322$, $\omega^2 = 0.2152$; Fig. 5e], or the basolateral amygdala [interaction: $F_{(1,13)} = 1.011$, $p = 0.3330$, $\omega^2 = 7.189$; main effect of cocaine: $F_{(1,13)} = 0.06364$, $p = 0.8048$, $\omega^2 = 0.4525$; main effect of HIV: $F_{(1,13)} = 0.0269$, $p = 0.8722$, $\omega^2 = 0.1914$; Fig. 5f].

### Cocaine exposure increased microglia number in the hippocampus of HIV-infected mice

To assess whether HIV infection or cocaine exposure altered the number of microglia, brain tissue was stained for detection of Iba1 and the number of Iba1+ cells was analyzed.

The hippocampus was analyzed by the four distinct subregions, the CA1 [Fig. 6a], CA2, CA3, and dentate gyrus (DG). A two-way ANOVA revealed a significant interaction of cocaine exposure and HIV infection on the number of Iba1+ cells in the CA1 subregion of the hippocampus [$F_{(1,27)} = 4.702$, $p = 0.0391$, $\omega^2 = 12.74$; Fig. 6b]. Tukey's HSD test for multiple comparisons found that the mean number of Iba1+ cells was significantly increased in cocaine-exposed HIV-infected mice, compared to saline-only HIV-infected mice [$p = 0.0214$, 95% C.I. −566.2 to −36.24].

Iba1+ cell counts were not significantly different in the CA2, CA3, or DG subregions of the hippocampus (Supplementary Fig. 1a–c). Similarly, no changes in Iba1+ cell counts were observed in the basolateral amygdala [interaction: $F_{(1,25)} = 0.4168$, $p = 0.5244$, $\omega^2 = 1.522$; main effect of cocaine: $F_{(1,25)} = 1.317$, $p = 0.2620$, $\omega^2 = 4.809$; main effect of HIV: $F_{(1,25)} = 0.9903$, $p = 0.3292$, $\omega^2 = 3.617$; Fig. 6c], the nucleus accumbens [interaction: $F_{(1,27)} = 0.4037$, $p = 0.5306$, $\omega^2 = 1.464$; main effect of cocaine: $F_{(1,27)} = 0.1133$, $p = 0.7390$, $\omega^2 = 0.411$; main effect of HIV: $F_{(1,27)} = 0.02305$, $p = 0.8805$, $\omega^2 = 0.08361$; Fig. 6d], the prelimbic PFC

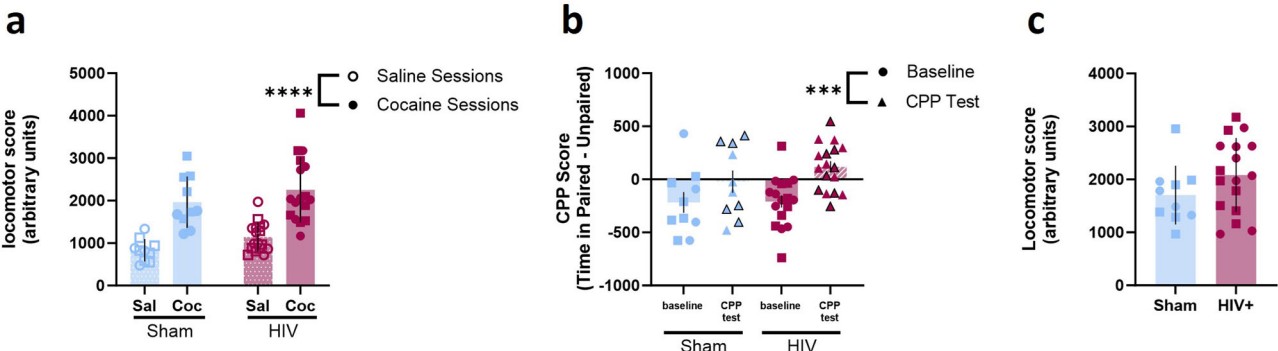

**Fig. 2 | Normal cocaine-induced locomotion and conditioned place preference in HIV-1 infected mice. a** Average locomotor scores of cocaine-exposed sham (n = 10) or HIV-infected (n = 17) mice during saline or cocaine-paired conditioning sessions. **b** No differences in CPP scores were observed in cocaine-exposed sham or HIV-infected mice. **c** No differences in locomotor score were observed during the CPP test in cocaine-exposed sham or HIV-infected mice. ***$p < 0.001$, ****$p < 0.0001$. Bars represent mean ± SEM. Squares and black-outlined triangles represent male mice, circles and solid symbols represent female mice. Open and closed symbols represent saline and cocaine exposed mice, respectively.

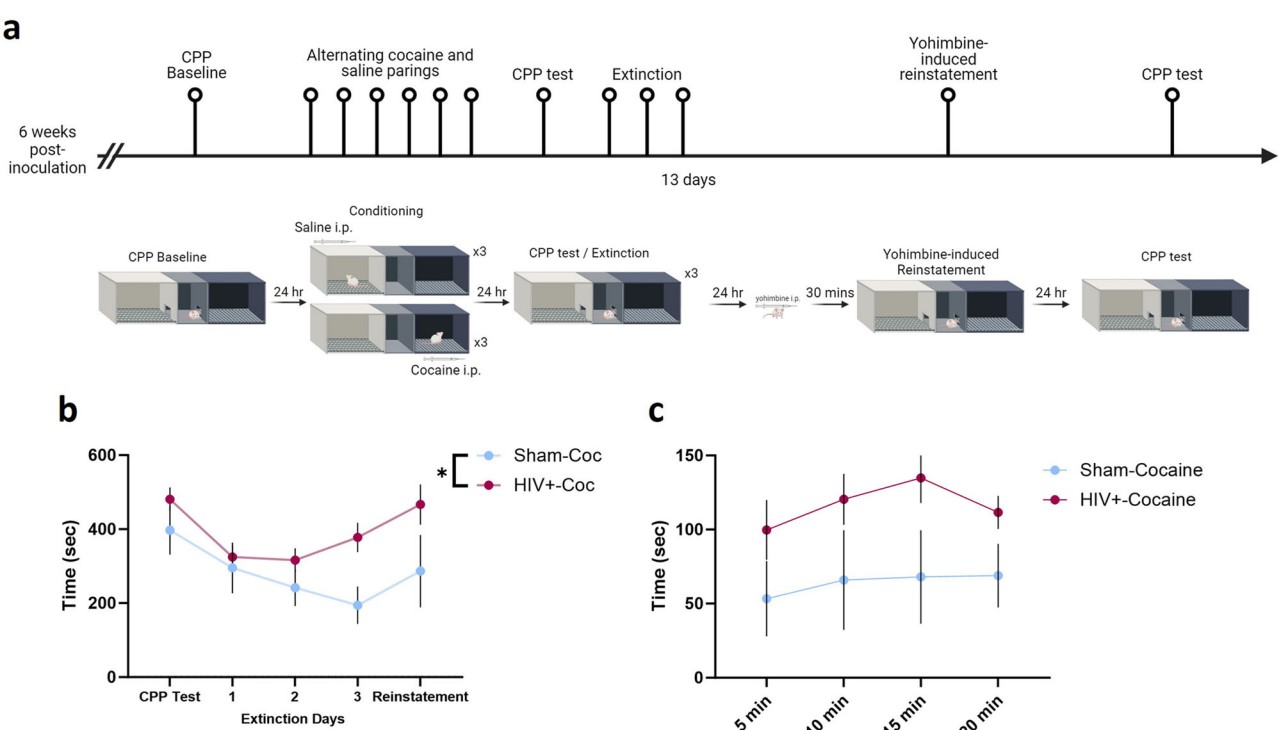

**Fig. 3 | Impaired extinction of cocaine CPP in HIV-1 infected mice. a** Timeline of CPP followed by extinction and reinstatement testing (created with Biorender.com). **b** A subset of mice underwent extinction of cocaine CPP following conditioning. HIV-1 infected mice exhibited persistent CPP across extinction sessions, indicative of a deficit in extinction. **c** Sham (n = 10) and HIV-1-infected (n = 16) mice exhibited similar magnitudes of yohimbine-induced reinstatement. *$p < 0.05$. Data represent mean ± SEM.

[interaction: F $(1,22) = 0.4181$, $p = 0.5246$, $\omega^2 = 1.859$; main effect of cocaine: F $(1,22) = 0.04738$, $p = 0.8297$, $\omega^2 = 0.2107$; main effect of HIV: F $(1,22) = 0.005112$, $p = 0.9436$, $\omega^2 = 0.02273$; Fig. 6e], or the infralimbic PFC [interaction: F $(1,22) = 0.3847$, $p = 0.5415$, $\omega^2 = 1.691$; main effect of cocaine: F $(1,22) = 0.0002616$, $p = 0.9872$, $\omega^2 = 0.001150$; main effect of HIV: F $(1,22) = 0.1507$, $p = 0.7016$, $\omega^2 = 0.6621$; Fig. 6f].

### Microglial activation is elevated in cocaine-exposed HIV-infected mice

To determine if microglial activation is changed by cocaine exposure or HIV infection, the percent area of Iba1+ cells was analyzed (Fig. 7a). A two-way ANOVA showed a main effect of cocaine on the percent area of Iba1+ cells in the basolateral amygdala [F $(1,25) = 4.468$, $p = 0.0447$, $\omega^2 = 13.98$;

Fig. 7b]. No effect of HIV [F $(1,25) = 0.5165$, $p = 0.4790$, $\omega^2 = 1.616$] or interaction [F $(1,25) = 0.5651$, $p = 0.4592$, $\omega^2 = 1.768$] was observed.

No changes in percent area of Iba1+ cells were observed in the CA1 [Interaction: F $(1,27) = 3.152$, $p = 0.0871$, $\omega^2 = 10.06$; main effect of cocaine: F $(1,27) = 0.8341$, $p = 0.3692$, $\omega^2 = 2.663$; main effect of HIV: F $(1,27) = 0.2408$, $p = 0.6276$, $\omega^2 = 0.7687$; Fig. 7c], CA2, CA3, or DG sub-regions of the hippocampus (Supplementary Fig. 1d–f). Iba1+ percent area was unchanged in the nucleus accumbens [interaction: F $(1,27) = 0.4171$, $p = 0.5238$, $\omega^2 = 1.416$; main effect of cocaine: F $(1,27) = 2.074$, $p = 0.1613$, $\omega^2 = 7.041$; main effect of HIV: F $(1,27) = 0.1278$, $p = 0.7235$, $\omega^2 = 0.4339$; Fig. 7d], the prelimbic PFC [interaction: F $(1,22) = 0.004373$, $p = 0.9479$, $\omega^2 = 0.01614$; main effect of cocaine: F $(1,22) = 1.944$, $p = 0.1772$, $\omega^2 = 7.176$; main effect of HIV: F $(1,22) = 2.170$, $p = 0.1549$,

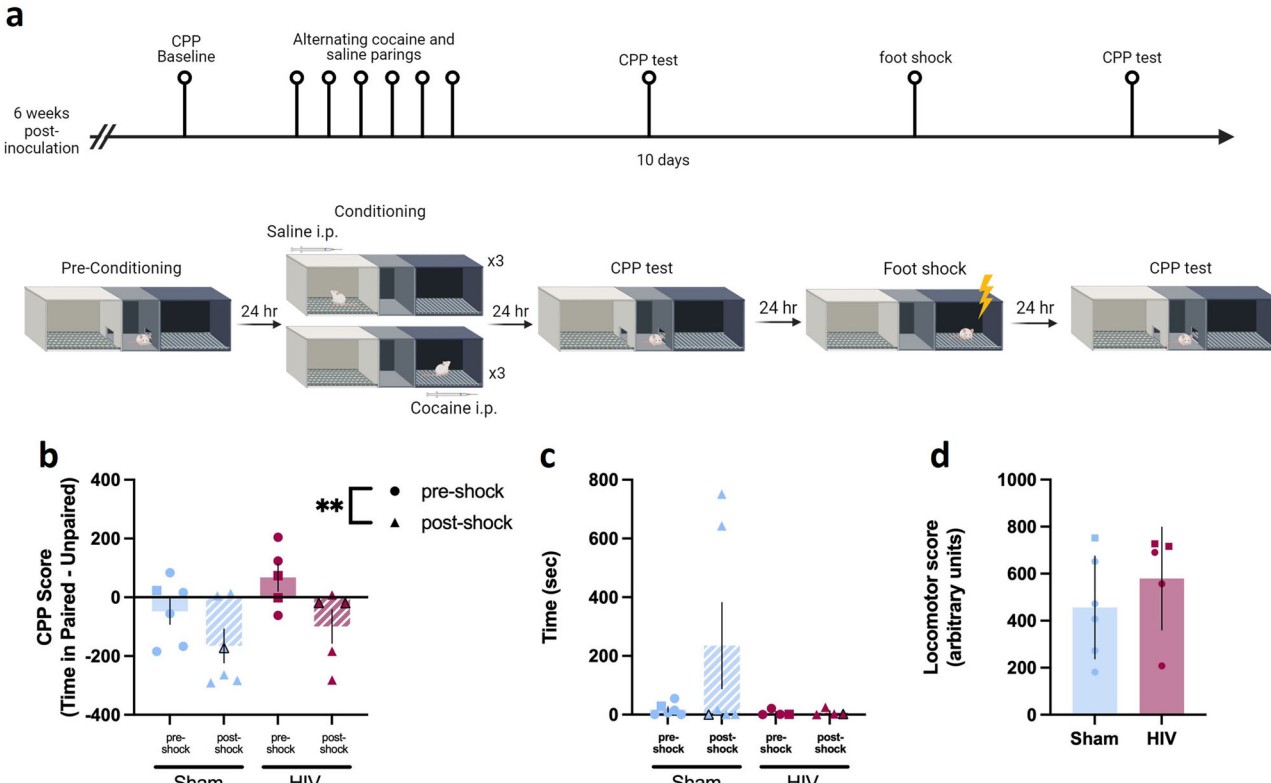

**Fig. 4 | HIV infection did not impact cocaine seeking under conflict. a** Timeline of CPP with footshock and testing for cocaine seeking under conflict (created with Biorender.com). **b** Sham (n = 6) and HIV-infected mice (n = 5) showed similar reductions of time spent in the cocaine-paired chamber when cocaine-footshock conflict was introduced. **c** Neither sham (n = 6) nor HIV-infected mice (n = 4) showed significant increases in latencies to enter the cocaine-paired chamber following the introduction of cocaine-footshock conflict. **d** No differences in loco-motor score were observed during the CPP test in cocaine-exposed sham (n = 6) or HIV-infected (n = 6) mice. **p < 0.01. Data represent mean ± SEM. Squares and black-outlined triangles represent male mice, circles and solid symbols represent female mice.

$\omega^2 = 8.011$; Fig. 7e], and infralimbic PFC [interaction: F (1,22) = 0.0391, $p = 0.8450$, $\omega^2 = 0.1426$; main effect of cocaine: F (1,22) = 2.093, $p = 0.1621$, $\omega^2 = 7.863$; main effect of HIV: F (1,22) = 2.117, $p = 0.1598$, $\omega^2 = 7.656$; Fig. 7f].

### Cytokine/chemokine discovery assays

To assess markers of inflammation in the humanized HIV-1 model, this study utilized mouse (Supplementary Table 1) and human (Table 1; significant results indicated in bolded text) cytokine discovery assays. Results from the mouse cytokine assay did not survive false discovery rate corrections. The uncorrected $p$ values are reported in the supplement for reference.

In the human cytokine assay, two-way ANOVA revealed changes in inflammatory markers. HIV-1 infection reduced expression of IL-12p40 [main effect of HIV: F (1,29) = 11.01, $p = 0.0025$, $\omega^2 = 26.66$, q = 0.0083; main effect of cocaine: F (1,29) = 0.1830, $P = 0.6719$, $\omega^2 = 0.4432$; interaction: F (1,29) = 0.2539, $p = 0.6182$, $\omega^2 = 0.6148$; Fig. 8a], MDC [main effect of HIV: F (1,34) = 13454, $p = 0.0009$, $\omega^2 = 28.85$, q = 0.0060; main effect of cocaine: F (1,33) = 0.09288, $p = 0.7625$, $\omega^2 = 0.1992$; interaction: F (1,33) = 0.0003, $p = 0.9874$, $\omega^2 < 0.001$; Fig. 8b], MIP-1β [main effect of HIV: F (1,33) = 6.279, $p = 0.0173$, $\omega^2 = 14.87$, q = 0.0431; main effect of cocaine: F (1,33) = 1.582, $p = 0.2173$, $\omega^2 = 3.745$; interaction: F (1,33) = 1.446, $p = 0.2377$, $\omega^2 = 3.434$; Fig. 8c], PDGF-AA [main effect of HIV: F (1,29) = 7.237, $p = 0.0117$, $\omega^2 = 18.15$, q = 0.0333; main effect of cocaine: F (1,29) = 1.132, $p = 0.2899$, $\omega^2 = 2.915$; interaction: F (1,29) = 2.776, $p = 0.1064$, $\omega^2 = 6.964$; Fig. 8d], and TNFα [main effect of HIV: F (1,33) = 11.74, $p = 0.0017$, $\omega^2 = 24.05$, q = 0.0083; main effect of cocaine: F (1,33) = 2.084, $p = 0.1583$, $\omega^2 = 4.268$; interaction F (1,33) = 2.086, $p = 0.1581$, $\omega^2 = 4.273$; Fig. 8e]. IFNα2 levels were increased in HIV-1 infected mice compared to controls [main effect of HIV: F (1,33) = 10.86, $p = 0.0024$, $\omega^2 = 23.68$, q = 0.0083; main effect of cocaine: F (1,33) = 1.166, $p = 0.2881$, $\omega^2 = 2.543$; interaction: F (1,33) = 1.122, $p = 0.2971$, $\omega^2 = 2.448$; Fig. 8f]. Interactions were observed between HIV infection and cocaine exposure in human IFNγ and GM-CSF. IFNγ was higher in sham mice exposed to cocaine compared to control sham mice [$p = 0.0017$], and both groups of HIV-infected mice ($p$'s < 0.0001; Fig. 9a). GM-CSF was increased in cocaine-exposed sham mice compared to all other groups ($p$'s < 0.001; Fig. 9b). There were not sufficient samples with detectable levels of RANTES/CCL5 across all groups for statistical comparison of means. However, the frequency of presence of detectable levels of RANTES/CCL5 was significantly different in HIV-1 infected mouse samples than sham mouse samples [$X^2$ (1, N = 38) = 11.7725, $p = 0.0006$], consistent with potentially reduced levels of RANTES/CCL5 in the HIV-1 group compared to the sham mice.

### Discussion

The current findings demonstrate that HIV-1 infection altered a subset of cocaine-related behaviors in mice, which was accompanied by peripheral inflammatory response and changes in brain microglia and astrocytes in neural substrates associated with cocaine reward and memory. HIV-1 infection did not impact acquisition of a cocaine CPP or reward seeking under conflict. In contrast, HIV-1 infected mice exhibited greater persistence of CPP under extinction conditions. Despite resistance to extinction of cocaine CPP, HIV-1 infected mice showed a similar magnitude of yohimbine-induced reinstatement to sham controls. These behavioral alterations were accompanied by changes in astrocyte immunoreactivity in the PFC such that HIV-1 infected mice showed greater GFAP staining in the IL and PL cortices. In contrast to HIV-1 effects on astrocytes, cocaine-induced alterations in microglia were observed.

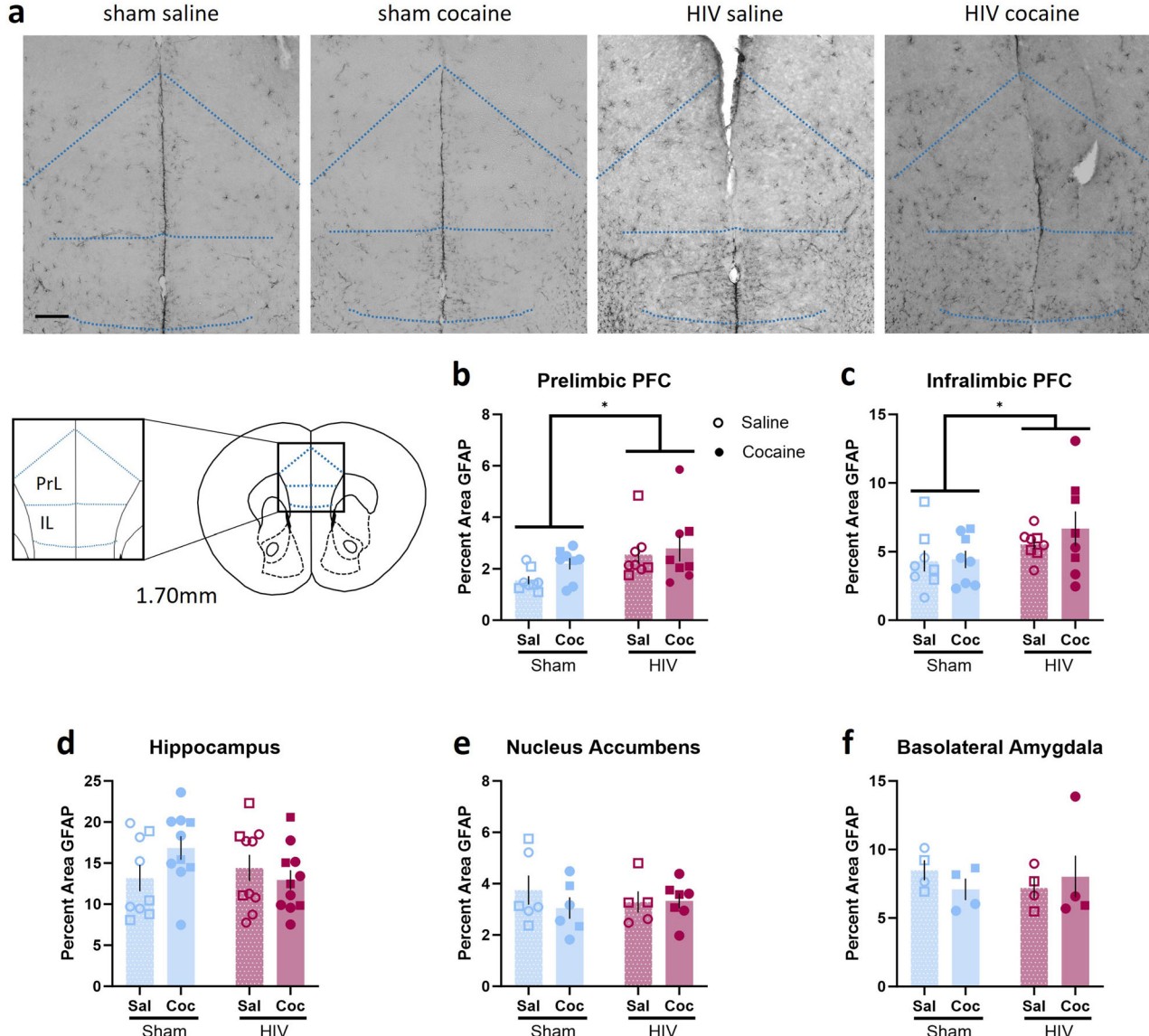

**Fig. 5 | HIV-1 infection altered GFAP immunoreactivity in reward-related neural substrates. a** Representative 20X images of GFAP staining in the PFC from sham-saline, sham-cocaine, HIV+-saline, and HIV+-cocaine mice. **b, c** HIV-1 infected mice exhibited increased GFAP immunoreactivity in the prelimbic (n = 8) and infralimbic (n = 8) subregions of the PFC. No effects of HIV-1 infection or cocaine exposure on GFAP immunoreactivity in the (**d**) hippocampus (Sham: n = 9 sal, 10 coc; HIV: n = 10 sal, coc), (**e**) nucleus accumbens (Sham: n = 6/group; HIV: n = 5 sal, 7 coc), or (**f**) basolateral amygdala (Sham: n = 4/group; HIV n = 4 sal, 5 coc) were observed. *$p < 0.05$. Data represent mean ± SEM. Scale bar represents 250 μm. Square and circle symbols represent male and female mice, respectively.

HIV-1 infected humanized mice exhibited normal formation of a cocaine conditioned place preference, which assesses cocaine-reward associations with discrete contexts. This suggests that HIV-infected mice did not exhibit gross impairments in cocaine reward-related learning and did not have profound differences in cocaine reward-related behaviors compared to the sham-inoculated controls. Although initial CPP was similar, HIV-1 infected mice exhibited persistent expression of cocaine CPP across extinction sessions. Persistent responding in the CPP task may also reflect greater reward-context conditioning in the initial CPP training that was not reflected in the CPP scores, or alternatively, differences in cocaine-seeking motivation. However, deficits in extinction of morphine CPP have been observed in the HIV-1 transgenic rat which expresses 7 of 9 HIV-1 viral proteins[26], though this model reflects consequences of protein exposure rather than infection. This may suggest generalized impairments in extinction across classes of rewards, though opioids likely interact with HIV to dysregulate cognitive function through discrete mechanisms from psychostimulants[27–30]. Although the magnitude of reinstatement did not differ, deficits in extinction in HIV-1 infected mice were reflected in the reinstatement session, with overall greater time in the reward-paired chamber in HIV-1 infected mice than in sham counterparts. This suggests that although the effects of peripheral HIV-1 infection on cocaine related behaviors were relatively modest in this paradigm, deficits in extinction resulted in overall greater persistence in cocaine seeking which could reflect difficulties in attaining and maintaining abstinence.

The ability to extinguish reward seeking is dependent on the infra-limbic cortex and its projections to the nucleus accumbens[31–33]. Deficits in extinction are particularly of note as our results identified increased GFAP immunoreactivity in the infralimbic and prelimbic cortices of HIV-infected mice, suggesting that prefrontal cortex function may be impaired. Indeed, others have found that astrocyte dysregulation in the medial prefrontal cortex is associated with extinction impairments in mice[34,35]. This included reduced expression of the cystine-glutamate antiporter, xCT (ref. [34]). The cystine-glutamate antiporter is known to be upregulated by exposure to Tat, though it is not clear how this system is altered in the humanized mouse

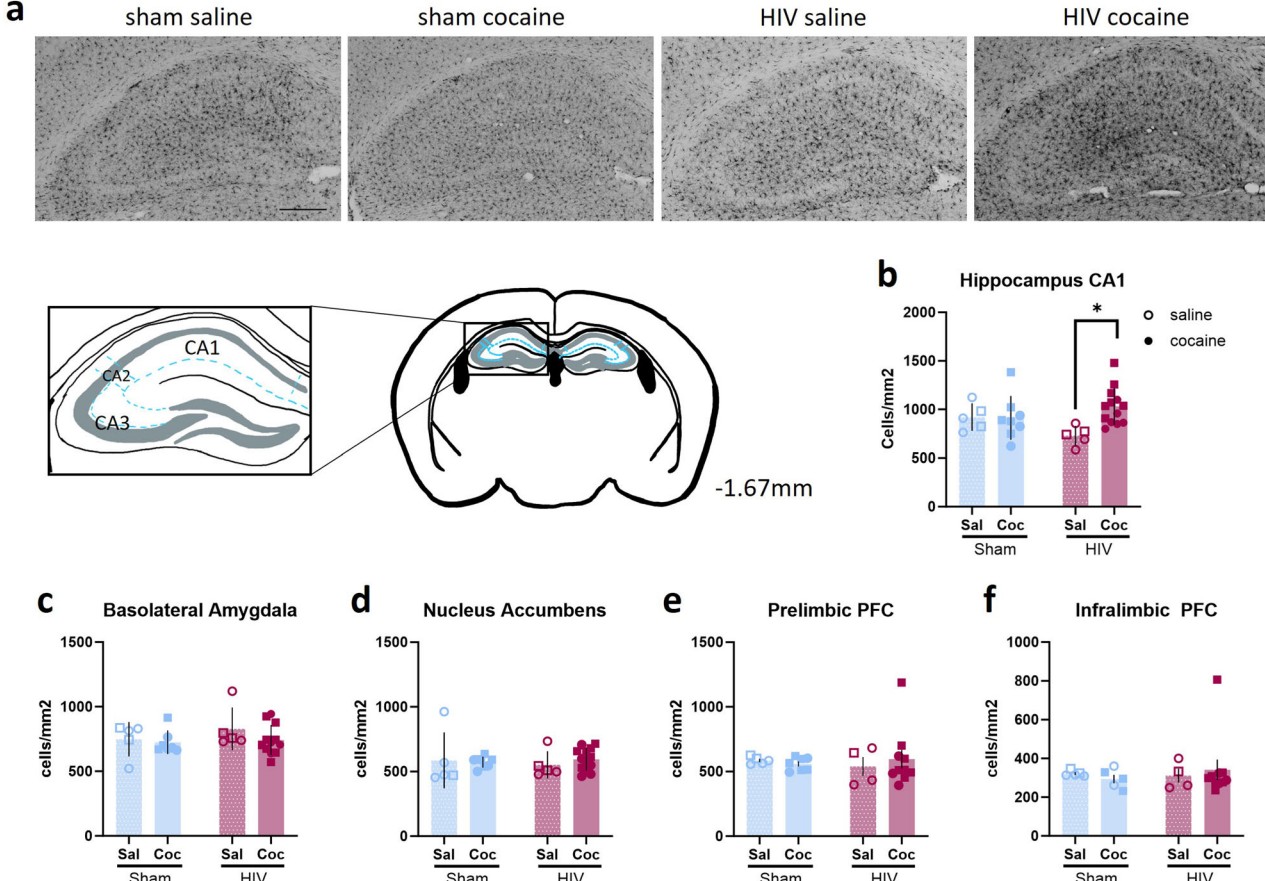

**Fig. 6 | HIV infection and cocaine exposure interacted to impact microglial number in the hippocampus. a** Representative 20X images of Iba1 staining in the hippocampus of sham-saline, sham-cocaine, HIV-saline, and HIV-cocaine mice. **b** A greater number of Iba1+ cells per mm² were observed in the CA1 subfield of the hippocampus in HIV-1 infected mice exposed to cocaine (Sham: n = 5 sal, 8 coc; HIV: n = 5 sal, 13 coc). No effect of cocaine was observed in sham mice. No effects of HIV-1 infection or cocaine exposure on microglia numbers were observed in the (**c**) basolateral amygdala (Sham: n = 5 sal, 7 coc; HIV: n = 5 sal, 12 coc), (**d**) nucleus accumbens (Sham: n = 5 sal, 8 coc; HIV: n = 5 sal, 13 coc), (**e**) prelimbic PFC (Sham: n = 5 sal, 7 coc; HIV: n = 4 sal, n = 10 coc), or (**f**) infralimbic (Sham: n = 5 sal, 7 coc; HIV: n = 4 sal, 10 coc) PFC. *$p < 0.05$. Data represent mean ± SEM. Scale bar represents 250 μm. Square and circle symbols represent male and female mice, respectively.

model. Both Tat and cocaine exposure independently alter astrocyte metabolism, resulting in reduced lactate transport to neurons, suggesting HIV-1 proteins may drive impairments in neuronal function indirectly through dysregulation of astrocytes[36]. We and others speculate that targeting astrocytes as regulators of dysregulated reward seeking behavior will be a promising therapeutic strategy across classes of addictive drugs[37–40]. The current findings suggest that astrocyte dysregulation is common across models of HIV, including infection and protein expression models, suggesting that astrocyte function and structure may further be promising for reversing HIV-associated changes in drug seeking behavior.

Yohimbine-induced reinstatement was used as a model of stress-induced relapse of drug seeking. Yohimbine produces physiological stress in humans and animals and is demonstrated to induce reinstatement of drug craving in clinical populations[41] and seeking in animal models[42–44]. Despite overall extinction-resistance in HIV-1 infected mice, the magnitude of yohimbine-induced reinstatement was similar in the HIV-1 infected mice and sham-infected counterparts, suggesting that stress-induced relapse-related behavior is not facilitated in this model. Interestingly, in the Tat Tg models, Tat expression facilitated the reinstatement of ethanol CPP[45], while the magnitude of cocaine-primed reinstatement was not altered in Tat Tg mice[14]. This may suggest that reinstatement of cocaine seeking, a model of relapse-related behavior, is relatively spared from the effects of HIV infection or protein exposure in both models of pharmacological stressor-induced (yohimbine) and in drug-primed reinstatement.

Previous research into cocaine-related behavior in preclinical models of HIV has been largely limited to males. Our finding showing similar magnitude of CPP may result from differential effects of HIV-1 on cocaine preference in this model as compared to the use of Tat transgenic animals[16], however, these previous findings were restricted to male mice. Substantial sex differences in cocaine-related behaviors have been reported, including development of cocaine CPP at lower doses in females[46] and reinstatement in females[47], and differences in cocaine preference in the EcoHIV model[48]. While the current study did not detect sex differences, a majority of the subjects were female mice. Others have found that in female mice, Tat-facilitation of cocaine CPP was restricted to the diestrus phase (Paris, Fenwick et al., 2014). The diestrus phase is associated with elevated progesterone levels in mice[49–53] which we have shown to reduce stress-induced reinstatement[43,54]. As Tat induction can dysregulate neurosteroid synthesis[55,56] and treatment with progesterone or its metabolite allopregnanolone can attenuate both behavioral and neuropathological outcomes of Tat induction[55,57], neurosteroids are implicated in both the development and treatment of neurobehavioral impairments in HIV. Future studies into the relationship between HIV infection and behavioral outcomes should consider hormonal status as a potential mediator of these outcomes. As astrocytes in the prefrontal cortex – a key substrate of reinstatement – exhibited increased immunoreactivity in HIV-1 infected mice in this model, and they further act as key regulators of steroid signaling in the brain[58,59], this further underscores the importance of considering circulating hormones and non-neuronal contributors in neuroHIV outcomes.

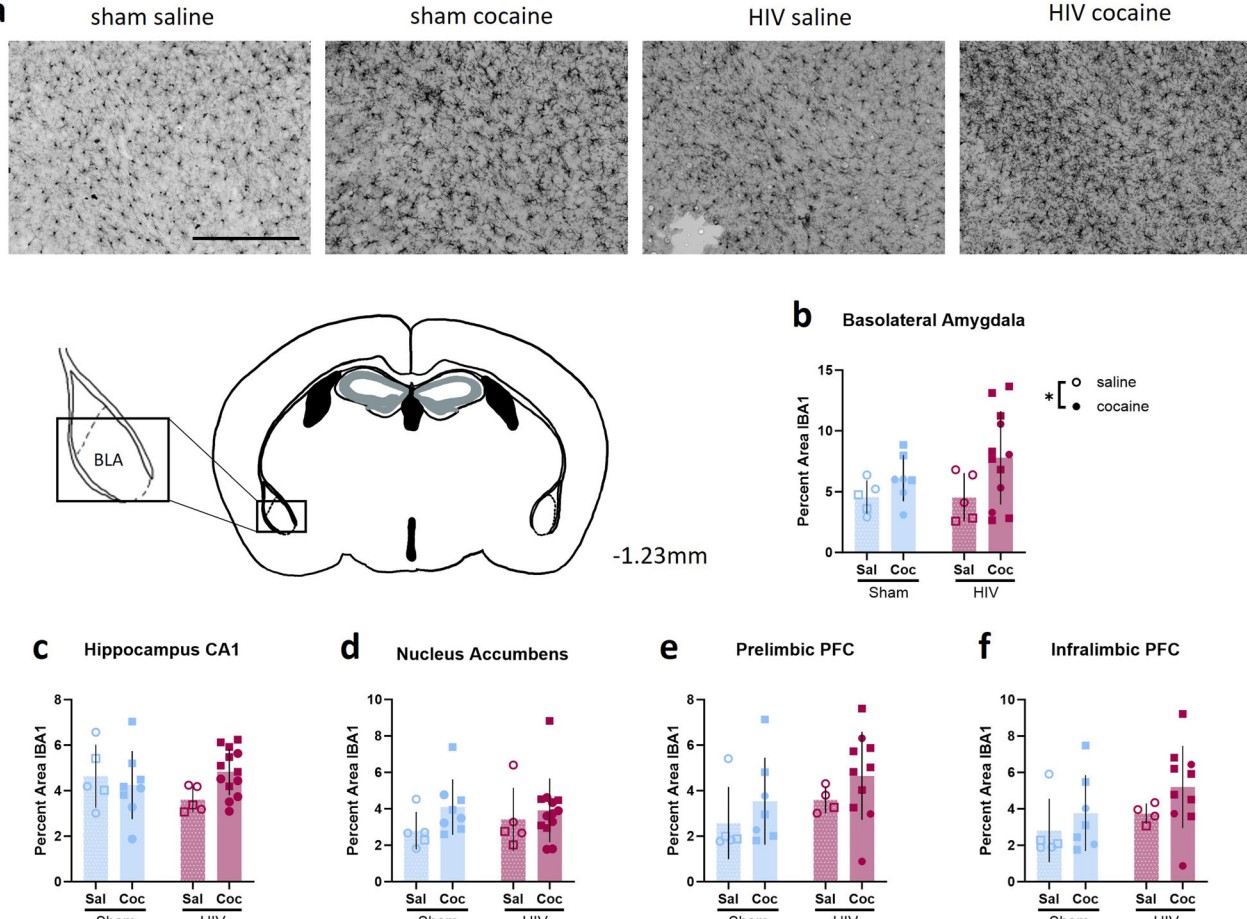

**Fig. 7 | Cocaine altered microglial activation in the basolateral amygdala.**
**a** Representative 20X images of Iba1 staining in the BLA from sham-saline, sham-cocaine, HIV-saline, and HIV-cocaine mice. **b** Independent of HIV-1 infection status, cocaine exposure increased the percent area of IBA1 staining in the BLA (Sham: n = 5 sal, 7 coc; HIV: n = 5 sal, 12 coc). No effects of HIV-1 infection or cocaine exposure were observed in the (**c**) nucleus accumbens (Sham: n = 5 sal, 8 coc; HIV: n = 5 sal, 13 coc), (**d**) CA1 (Sham: n = 5 sal, 8 coc; HIV: n = 5 sal, 3 coc) region of the hippocampus, (**e**) prelimbic PFC, or (**f**) infralimbic PFC (Sham: n = 5 sal, 7 coc; HIV: n = 4 sal, 10 coc). *$p < 0.05$. Data represent mean ± SEM. Scale bar represents 250 µm. Square and circle symbols represent male and female mice, respectively.

In contrast to prefrontal astrocyte changes induced by HIV-1 infection, alterations in microglia populations induced by cocaine exposure were observed in the basolateral amygdala and the CA1 subfield of the hippocampus. While cocaine exposure increased overall Iba1 staining in the basolateral amygdala - indicative of cocaine-induced morphology changes consistent with activated microglia in HIV-1 infected and uninfected mice, cocaine only increased microglia cell counts in the hippocampus of HIV-1 infected mice. This represents a unique vulnerability of the hippocampus to microglial dysregulation by cocaine in the context of HIV-1. The hippocampus is essential for normal learning and memory processes, including extinction of cocaine conditioned place preference[60], which was impaired in HIV-1 infected mice. These findings suggest a different profile of hippocampal microglia response to systemic HIV-1 infection when compared to local infusion of HIV-1 proteins to the hippocampus, which induces microglial activation[61]. This may reflect higher exposure to HIV-1 proteins resulting from this local infusion to the tissue. Hippocampal vulnerability in the EcoHIV model of systemic infection has also been reported[62,63], which is accompanied by altered hippocampus synaptodendritic integrity. Hippocampus network function has also been shown to be dysregulated by both exposure to HIV-1 Tat and cocaine[64]. Notably, cocaine and Tat-induced deficits in hippocampus network activity were attenuated in the presence of astrocytes. While we did not observe changes in GFAP immunoreactivity in the hippocampus of HIV-1 infected mice, these blunt measures may miss changes in astrocytic synaptic coverage or function which could contribute to changes in hippocampal synaptodendritic integrity or plasticity when

exposed to cocaine or elevated dopamine levels, resulting in aberrant hippocampal function. Importantly, although the morphology of Iba1-labeled cells was consistent with microglia, Iba1 can also label macrophages. As HIV infection may increase peripheral immune cell migration into the central nervous system, it is possible that macrophage populations contribute to observed effects of HIV on Iba1 expressing populations. Because this model includes humanization of the peripheral immune system, changes in the CNS to astrocyte and microglia populations as a result of HIV-1 infection likely in part reflect outcomes of peripheral infection and underscore the effects of inflammation on the brain and behavior.

These neural changes were accompanied by inflammatory changes in peripheral blood following behavioral testing, particularly in human cytokines and chemokines. Human granulocyte-macrophage colony stimulating factor (GM-CSF) levels were increased by cocaine exposure in uninfected mice, while this effect was blunted in mice infected with HIV-1. This aligns with previous findings that in a cocaine sensitization model, repeated cocaine exposure increased interferon gamma levels in immunointact mice[65]. Further, cocaine-induced increases in GM-CSF have been observed in other models of experimenter administered cocaine[66] under conditions that similarly induced increases in G-CSF. In PLWH who use drugs, GM-CSF levels were significantly associated with historical cocaine use[67], identifying dysregulation of GM-CSF as a common target in both preclinical and clinical data sets. Notably, human targets that were modulated by cocaine were generally pro-inflammatory, with cocaine-induced increases in sham mice in interferon gamma and GM-CSF. Consistent with

**Table 1 | Values of Analytes detected by Human Cytokine/Chemokine Panel A 48-Plex Discovery Assay**

| Analyte (units) | Abbreviation | Effect of HIV | Effect of Cocaine | Interaction |
|---|---|---|---|---|
| Fibroblast growth factor 2 | FGF-2 | $F_{(1, 32)} = 0.4247$, $P = 0.5192$ | $F_{(1, 32)} = 0.3991$, $P = 0.5320$ | $F_{(1, 32)} = 1.375$, $P = 0.2497$ |
| Granulocyte-macrophage colony-stimulating factor | GM-CSF | **$F_{(1, 33)} = 19.94$, $P < 0.0001$** | **$F_{(1, 33)} = 9.377$, $P = 0.0043$** | **$F_{(1, 33)} = 11.68$, $P = 0.0017$** |
| Human interferon alpha-2 | IFNα2 | **$F_{(1, 33)} = 10.86$, $P = 0.0024$** | $F_{(1, 33)} = 1.166$, $P = 0.2881$ | $F_{(1, 33)} = 1.122$, $P = 0.2971$ |
| Interferon gamma | IFNγ | **$F_{(1, 32)} = 18.93$, $P = 0.0001$** | **$F_{(1, 32)} = 7.930$, $P = 0.0083$** | **$F_{(1, 32)} = 8.373$, $P = 0.0068$** |
| Interferon gamma inducible protein-10 | IP-10 | $F_{(1, 33)} = 3.848$, $P = 0.0583$ | $F_{(1, 33)} = 0.02014$, $P = 0.8880$ | $F_{(1, 33)} = 0.2940$, $P = 0.5913$ |
| Interleukin 1-alpha | IL-1α | $F_{(1, 33)} = 1.916$, $P = 0.1756$ | $F_{(1, 33)} = 0.7734$, $P = 0.3855$ | $F_{(1, 33)} = 0.09463$, $P = 0.7603$ |
| Interleukin 2 | IL-2 | $F_{(1, 26)} = 2.096$, $P = 0.1596$ | $F_{(1, 26)} = 0.08028$, $P = 0.7792$ | $F_{(1, 26)} = 0.5794$, $P = 0.4534$ |
| Interleukin 3 | IL-3 | $F_{(1, 32)} = 0.3023$, $P = 0.5863$ | $F_{(1, 32)} = 2.226$, $P = 0.1455$ | $F_{(1, 32)} = 1.379$, $P = 0.2490$ |
| Interleukin 4 | IL-4 | $F_{(1, 32)} = 0.2148$, $P = 0.6462$ | $F_{(1, 32)} = 0.2510$, $P = 0.6198$ | $F_{(1, 32)} = 0.7015$, $P = 0.4085$ |
| Interleukin 5 | IL-5 | $F_{(1, 32)} = 3.680$, $P = 0.0640$ | $F_{(1, 32)} = 3.259$, $P = 0.0805$ | $F_{(1, 32)} = 0.4367$, $P = 0.5134$ |
| Interleukin 6 | IL-6 | $F_{(1, 33)} = 3.377$, $P = 0.0751$ | $F_{(1, 33)} = 1.150$, $P = 0.2913$ | $F_{(1, 33)} = 1.057$, $P = 0.3114$ |
| Interleukin 7 | IL-7 | $F_{(1, 33)} = 0.2014$, $P = 0.6566$ | $F_{(1, 33)} = 0.1989$, $P = 0.6585$ | $F_{(1, 33)} = 0.1421$, $P = 0.7086$ |
| Interleukin 12 subunit-beta | IL-12p40 | **$F_{(1, 29)} = 11.01$, $P = 0.0025$** | $F_{(1, 29)} = 0.1830$, $P = 0.6719$ | $F_{(1, 29)} = 0.2539$, $P = 0.6182$ |
| Interleukin 13 | IL-13 | $F_{(1, 28)} = 2.737$, $P = 0.1092$ | $F_{(1, 28)} = 0.1380$, $P = 0.7131$ | $F_{(1, 28)} = 1.589$, $P = 0.2178$ |
| Interleukin 22 | IL-22 | $F_{(1, 32)} = 1.595$, $P = 0.2157$ | $F_{(1, 32)} = 1.832$, $P = 0.1853$ | $F_{(1, 32)} = 0.05059$, $P = 0.8235$ |
| Interleukin 25 | IL-17E/IL-25 | $F_{(1, 33)} = 0.7120$, $P = 0.4049$ | $F_{(1, 33)} = 0.0002755$, $P = 0.9869$ | $F_{(1, 33)} = 0.8780$, $P = 0.3556$ |
| Interleukin 27 | IL-27 | $F_{(1, 28)} = 0.001979$, $P = 0.9648$ | $F_{(1, 28)} = 0.1803$, $P = 0.6743$ | $F_{(1, 28)} = 0.6510$, $P = 0.4266$ |
| Macrophage inflammatory protein-1 alpha | MIP-1α | $F_{(1, 33)} = 0.9331$, $P = 0.3411$ | $F_{(1, 33)} = 0.7257$, $P = 0.4004$ | $F_{(1, 33)} = 2.161$, $P = 0.1511$ |
| Macrophage inflammatory protein-1 beta | MIP-1β | **$F_{(1, 33)} = 6.279$, $P = 0.0173$** | $F_{(1, 33)} = 1.582$, $P = 0.2173$ | $F_{(1, 33)} = 1.446$, $P = 0.2377$ |
| Macrophage-derived chemokine | MDC | **$F_{(1, 33)} = 13.45$, $P = 0.0009$** | $F_{(1, 33)} = 0.09288$, $P = 0.7625$ | $F_{(1, 33)} = 0.0002535$, $P = 0.9874$ |
| Monocyte Chemoattractant Protein-1, now called chemokine ligand 2 (CCL2) | MCP-1 | $F_{(1, 27)} = 3.079$, $P = 0.0907$ | $F_{(1, 27)} = 0.9709$, $P = 0.3332$ | $F_{(1, 27)} = 1.058$, $P = 0.3128$ |
| Monocyte Chemoattractant Protein-3, now called chemokine ligand 7 (CCL7) | MCP-3 | $F_{(1, 32)} = 3.439$, $P = 0.0729$ | $F_{(1, 32)} = 1.019$, $P = 0.3204$ | $F_{(1, 32)} = 1.456$, $P = 0.2364$ |
| Monokine induced by gamma interferon, Chemokine ligand 9 | MIG/CXCL9 | $F_{(1, 31)} = 2.104$, $P = 0.1569$ | $F_{(1, 31)} = 0.9067$, $P = 0.3484$ | $F_{(1, 31)} = 0.8674$, $P = 0.3589$ |
| Platelet-derived growth factor AA | PDGF-AA | **$F_{(1, 29)} = 7.237$, $P = 0.0117$** | $F_{(1, 29)} = 1.162$, $P = 0.2899$ | $F_{(1, 29)} = 2.776$, $P = 0.1064$ |
| Platelet-derived growth factor AB/BB | PDGF-AB/BB | **$F_{(1, 33)} = 5.365$, $P = 0.0269$** | $F_{(1, 33)} = 0.6004$, $P = 0.4439$ | $F_{(1, 33)} = 0.6860$, $P = 0.4135$ |
| Tumor necrosis factor alpha | TNFα | **$F_{(1, 33)} = 11.74$, $P = 0.0017$** | $F_{(1, 33)} = 2.084$, $P = 0.1583$ | $F_{(1, 33)} = 2.086$, $P = 0.1581$ |

Results of the Human Cytokine Discovery Assay. Main effects of HIV and cocaine and interactions are represented by **bolded text**. The p values reported in this table represent uncorrected values, before the false discovery rate test was applied.

findings from the literature, this may implicate cocaine in producing a pro-inflammatory state and generalized impairment in immune function. The current findings reflect repeated, but not chronic, cocaine exposure, and thus it will be critical to expand these results to models of chronic drug exposure and dependence. Mice in these experiments underwent yohimbine-primed reinstatement several days prior to assessing cytokine levels. While this did not differ based on HIV- or cocaine-treatment status, it is possible that adrenergic signaling differentially impacted cytokine/chemokine expression outcomes as there is significant evidence of interplay between the adrenergic system and immune outcomes, including within the context of cocaine or HIV[68–71]. To our knowledge, a mechanistic role of GM-CSF and interferon gamma have not been assessed in the regulation of cocaine-related behavior either systemically or in reward-related neural structures, suggesting a potential avenue for reversal of HIV-1 associated alterations in cocaine reward-related behaviors. In these studies, changes in GM-CSF and interferon gamma were similar, in that both were increased by cocaine exposure in sham mice, but unmodulated by cocaine in HIV-infected mice. This may be consistent with a role for directional changes – with increases in interferon gamma driving GM-CSF upregulation, but the time course of our analyses cannot determine this relationship. Notably, both GM-CSF and interferon gamma in HIV-1 have been shown to have both stimulatory and inhibitory effects on HIV infection, though this may depend on model and treatment approach[72]. This may suggest that cocaine induced changes in GM-CSF and interferon gamma could impact subsequent risk for HIV infection or infection outcomes in people with a history of chronic cocaine exposure.

The current study provides additional understanding of the interaction between HIV infection and cocaine exposure to alter the immune landscape and promote continued drug seeking. This area of investigation has faced challenges due to difficulties in modeling HIV infection in preclinical models. This study used a model in which mice are humanized by a process in which the mouse's immune cells are irradiated at birth, and human HPSCs are engrafted. The humanized mouse model thus offers insight into a number of unique questions, including how HIV-1 infection per se as opposed to protein exposure alone, is impacting central and peripheral inflammation and impacting behavior. These findings have complemented findings from other models, including the ecoHIV model and Tat transgenic model, with several converging areas—such as extinction learning and neuroimmune function—, which we expect to be particularly robust targets for future investigation. Limitations of the current model included behavioral abnormalities such as high rates of stereotypy of mice and differences in overall brain morphology, which prevented automation of analyses and will require robust calibration for future work manipulating discrete neural circuits as neuroanatomical differences between the humanized mice and commonly used inbred mouse lines were the norm.

Together, findings in these studies demonstrate that, in mice with humanized immune systems, HIV-1 infection is associated with deficits in

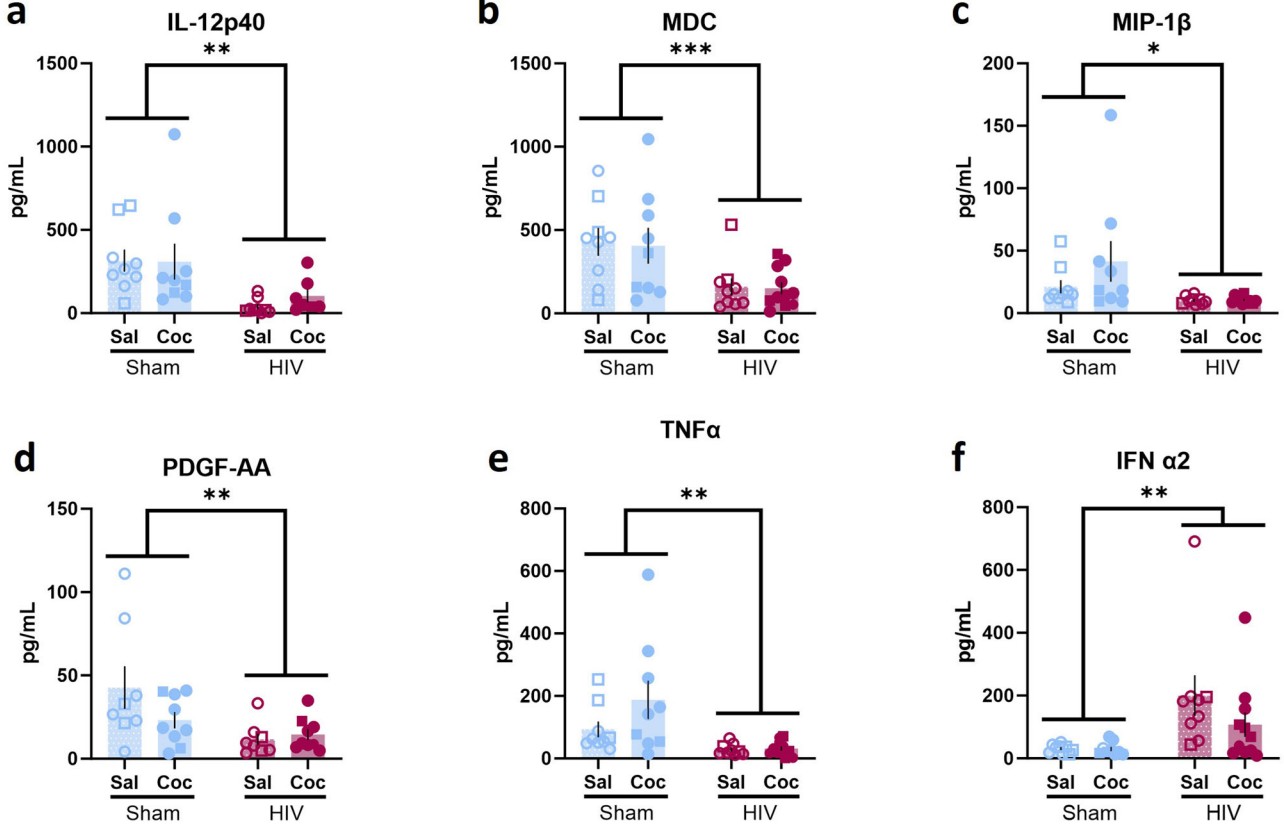

**Fig. 8 | HIV infection impacted human inflammatory factors in peripheral blood.** A main effect of HIV, but not cocaine, was observed in (**a**) IL-12p40 (Sham: n = 9/group; HIV: n = 9 sal, 7 coc), (**b**) MDC (Sham: n = 9/group; HIV: n = 9 sal, 11 coc), (**c**) MIP-1β (Sham: n = 9/group; HIV: n = 9 sal, 11 coc), (**d**) PDGF-AA (Sham: n = 8 sal, 9 coc; HIV: n = 8 sal, 9 coc), (**e**) TNFα (Sham: n = 9/group; HIV: n = 9 sal, 11 coc) and (**f**) IFN α2 (Sham: n = 9/group; HIV: n = 9 sal, 11 coc). *$p < 0.05$, ***$p < 0.001$, ****$p < 0.0001$. Data represent mean ± SEM. Square and circle symbols represent male and female mice, respectively.

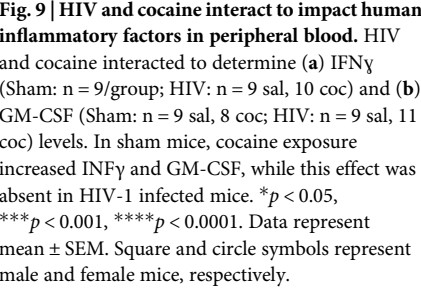

**Fig. 9 | HIV and cocaine interact to impact human inflammatory factors in peripheral blood.** HIV and cocaine interacted to determine (**a**) IFNγ (Sham: n = 9/group; HIV: n = 9 sal, 10 coc) and (**b**) GM-CSF (Sham: n = 9 sal, 8 coc; HIV: n = 9 sal, 11 coc) levels. In sham mice, cocaine exposure increased INFγ and GM-CSF, while this effect was absent in HIV-1 infected mice. *$p < 0.05$, ***$p < 0.001$, ****$p < 0.0001$. Data represent mean ± SEM. Square and circle symbols represent male and female mice, respectively.

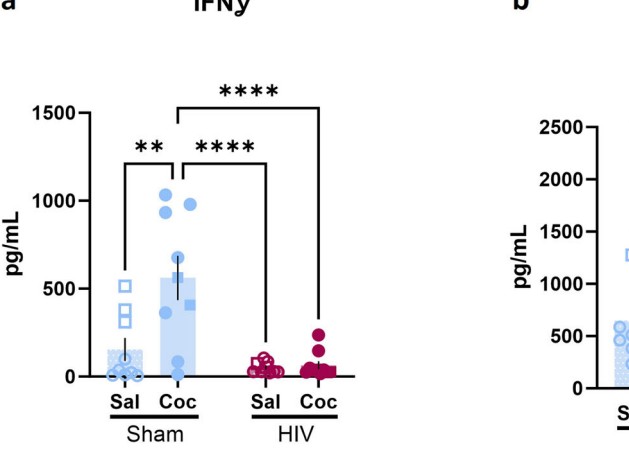

extinction of cocaine conditioned place preference. This is associated further with astrocyte and microglia alterations in key substrates of reward extinction, including subregions of the medial prefrontal cortices and the hippocampus. Further, systemic inflammatory alterations resulted from HIV-1 infection and cocaine exposure, both independently and in tandem, which identifies additional targets of investigation for peripheral immune consequences within the central nervous system that may contribute to deficits in the regulation of cocaine seeking behavior in the context of HIV-1 infection.

## Methods

### Subjects

Adult male (n = 25) and female (n = 40) NOD/Scid IL2Rg-/- (NSG) mice from the University of Nebraska Medical Center were used for these studies[73–77]. Prior to arriving at Drexel University Animal Facilities, transgenic NOD/Scid IL2Rg-/- (NOG) mice were irradiated and underwent transplantation of human hematopoietic stem and progenitor cells (HPSCs). Blood was collected and tested prior to transportation to ensure successful engraftment. The average age upon arrival was 182 days.

Following arrival, mice underwent a 2-week quarantine period, followed by placement in their final housing location under microisolation conditions at Drexel University's Center City campus (Fig. 1a). All mice were housed in same sex cages with *ad libitum* access to food and water in a temperature and humidity-controlled environment under a standard 12-h light/dark cycle. Mice were given supplementary diet gel as needed to maintain healthy body weights. Any mice that did not survive relevant behavioral testing or exhibited unusual grooming or circling behavior were removed from data analysis. All procedures were approved by the Institutional Animal Use and Care Committee at Drexel University. We have complied with all relevant ethical regulations for animal use.

### HIV inoculation and viral load testing

Following a 1-week acclimation, mice were matched based on sex and engraftment status into either HIV or sham control groups. Mice were inoculated with 250 ng p24 of the R5-tropic HIV-1 strain ADA (HIV$_{ADA}$). For inoculation, mice were briefly anesthetized with 5% vaporized isoflurane and injected interperitoneally with either HIV-1 or vehicle (RPMI medium) for sham controls prior to being returned to their home cage.

Blood was collected from all mice every 2 weeks. For blood collections, mice were briefly anesthetized with 5% vaporized isoflurane and, using the submandibular collection method, ~200 μL of blood was collected from each mouse and spun at 9000 rpm for 20 min at 4 °C. Plasma was collected and stored at −80 °C. Viral load for each mouse was determined via Q-RT-PCR performed by the Penn CFAR Viral Reservoir Core at the University of Pennsylvania. RNA was prepared using Tri reagent. Samples were run in duplicate using an LTR primer-probe to determine HIV-1 RNA copies. The sequence of primers for RCAS were 5′- GTC AAT AGA GAG AGG GAT GGA CAA A-3′ and R- 5′-TCC ACA AGT GTA GCA GAG CCC-3′ and the sequences for HIV-1 LTR were 5′-GCC TCA ATA AAG CTT GCC TTG A-3′ and R- 5′-GGG CGC CAC TGC TAG AGA-3′. RCAS virion was spiked in each sample and amplified separately to confirm virus/RNA recovery and a lack of PCR inhibition.

### Conditioned place preference paradigm

The CPP paradigm was used to investigate cocaine reward-related behaviors. All behavioral training and testing took place during the light cycle, between zeitgeber times 6 and 9, Monday through Friday. Med Associates CPP boxes were used for all experiments. These apparatuses consist of 3 distinct chambers, connected by openings which can be closed by the experimenter. The small center chamber had gray walls and a solid gray floor. On either side of the center chamber were two larger chambers, one with black walls with metal bar flooring, the other with white walls with a metal grid floor. On the first day of the CPP protocol, mice were placed in the center chamber and allowed to explore all 3 chambers freely for 20 min. Using Med Associates software, data collected included latency to enter each chamber, number of entries, total movement and activity, and time spent in each chamber. Data from this session were used as an initial baseline for preference. Mice were then matched based on infection status to either cocaine or saline-only control groups. Mice in the cocaine group received an i.p. injection of cocaine (10 mg/kg) immediately prior to placement in the assigned reward-paired chamber. This dose was selected as it consistently yields cocaine CPP in mice[78–80]. On alternating days, mice received a saline injection prior to placement in the saline-paired chamber. There was a total of 6 conditioning sessions (3 each, cocaine and saline pairings). Saline controls received saline prior to all sessions. The order of sessions was counterbalanced among groups for initial preference, drug condition, and infection status. To assess the development of a preference for the cocaine-paired chamber, mice were placed into the gray chamber and allowed to explore all 3 chambers freely for 20 (extinction and reinstatement tasks) or 5 min (compulsive-like cocaine seeking task).

**Extinction and reinstatement.** In order to investigate persistent expression of a CPP after the cocaine-context relationship was removed, one cohort of mice underwent extinction training. In the 3 consecutive extinction sessions, mice were placed in the center chamber and allowed to freely explore all 3 chambers for 20 min in the absence of cocaine. To investigate a model of stress-primed reinstatement of cocaine seeking in HIV infection, the ability of a pharmacological stressor, yohimbine, to reinstate cocaine seeking was assessed. One day after the third extinction session, mice were administered a single dose of the α2 adrenergic antagonist yohimbine (2 mg/kg) 30 min prior to placement in the CPP apparatus. This dose was selected as it has previously been demonstrated to yield reinstatement to cocaine CPP in mice[81]. These sessions were identical to the extinction sessions other than drug treatment. One mouse in this cohort died following CPP testing, prior to the beginning of extinction training.

**Cocaine seeking under conflict.** To investigate persistent cocaine-seeking despite adverse consequences, a separate cohort of mice experienced a footshock in the cocaine-paired chamber following conditioning under parameters that have been shown to suppress alcohol[82,83] and food[84] seeking behaviors in mice. Briefly, the CPP test consisted of a 5-min session to avoid extinction of the learned cocaine-chamber association. 24 h after the CPP test, mice were restricted to the cocaine-paired chamber for 3 min and received a mild (0.8 mA, 2 s) footshock 1 min after placement. The following day mice underwent a 20-min CPP test session, and the first 5 min were quantified to compare to the pre-shock test session.

### Brain tissue processing and immunohistochemistry

Following the conclusion of all behavioral testing, mice were anesthetized, and blood was collected for viral load testing via submandibular bleeds for comparison to other time points. Mice were then overdosed on isoflurane and transcardially perfused with 1X PBS solution followed by 4% Paraformaldehyde (PFA). Brains were collected and stored in 4% PFA for 24 h, then cryoprotected in sucrose solution for subsequent immunohistological processing for assessment of astrocyte immunoreactivity via GFAP immunohistochemistry and putative microglia number and reactivity via Iba1 immunohistochemistry.

Using a cryostat, 40 μm coronal sections were collected in quadruplicate and stored in 0.01% sodium azide. Tissue sections were incubated in 1% hydrogen peroxide for 1 h, then blocked in 5% normal donkey serum for 1 h prior to incubation in primary antibody (anti-GFAP primary antibody, 1:10,000, G9269, Sigma-Aldrich or anti-Iba1 primary antibody 1:10,000, 019-19741, Wako) overnight at room temperature. Sections were then incubated in biotinylated donkey-anti-rabbit secondary (1:1,000, 711-065-152, Jackson labs) for 30 min. Staining was visualized using DAB for GFAP staining and nickel-enhanced DAB (Vector Labs, SK-4100) for Iba1 staining for 20 min. Sections were mounted on plus slides and coverslipped with DPX mounting medium.

For both GFAP and Iba1 analysis, 10X images were captured and stitched together using Microsoft Image Composite Editor. Images were analyzed using ImageJ software, using a thresholding approach. Number of cells (cells/area) and percent area stained were analyzed for the prefrontal cortex (PFC, 1.53 mm, 1.7 mm, and 1.97 mm anterior of bregma), nucleus accumbens core and shell (1.54 mm, 1.34 mm, 1.1 mm, and 0.98 mm anterior of bregma), the basolateral amygdala (−0.95 mm, −1.07 mm, −1.23 mm, and −1.31 mm posterior of bregma) and dorsal hippocampus subregions (−1.3 mm, −1.46 mm, −2.2 mm, and −2.9 mm posterior of bregma) by an investigator blind to conditions.

### Cytokine and chemokine assessment

This study used the Luminex xMAP technology for multiplexed quantification of 48 human cytokines, chemokines, and growth factors and 32 mouse cytokines, chemokines, and growth factors. The multiplexing analysis was performed using the Luminex 200 system (Luminex, Austin, Texas) by Eve Technologies Corporation (Calgary, Alberta). Forty-eight human markers and 32 mouse markers were measured in plasma samples from week 7 or at the time of euthanasia using Eve Technologies' Human Cytokine Panel A

48-Plex Discovery Assay and Mouse Cytokine, 32-Plex Discovery Assay (Millipore Sigma, Burlington, Massachusetts) according to the manufacturer's protocol. Assay sensitivities of the human markers range from 0.14 to 50.78 pg/mL and the mouse markers range from 0.3 to 30.6 pg/mL. Markers included in the analysis are reported in Table 1 and Supplementary Table 1. A subset of targets was excluded because fewer than 5 samples per group were within the range of detection or within the standard curve.

## Statistics and reproducibility

All data were analyzed in GraphPad. Behavioral data were analyzed by repeated measures ANOVA for 2-way comparisons or by unpaired t-tests. Cytokine/chemokine expression data and immunohistological data were analyzed with 2way ANOVA. False discovery rate correction was applied to results from array data to maintain alpha at 0.05 using the two-stage linear step-up Benjamini, Krieger, Yuketeli procedure. Analyses were Greenhouse-Geisser corrected when sphericity was not met. Significant interactions were followed with post hoc analyses. All source data in this manuscript are available in the Supplementary Materials. Behavioral data was collected and analyzed across three cohorts of mice. Cytokine and chemokine arrays were run in duplicate. Immunohistochemistry data reflect the mean of each animal across 3–4 brain sections.

## Reporting summary

Further information on research design is available in the Nature Portfolio Reporting Summary linked to this article.

## Data availability

Numerical source data for all figures are provided in the supplement accompanying this manuscript (Supplementary Data File 1).

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

## Acknowledgements
This research was supported by NIH awards DP2DA051907 (J.M.B.), R03DA047919 (J.M.B.), 1R01DA054535-01 (S.G.), and 1R33DA041018-01 (S.G.), and a pilot award from The Comprehensive Neuro-AIDS Center Grant P30MH092177-9 (J.M.B. and L.L.G.).

## Author contributions
J.M.B designed and supervised the study, and prepared the paper, L.B. conducted experiments and prepared the paper, Q.X., M.W., and C.M.S. conducted immunohistochemistry analysis, L.L.G. conducted immunohistochemistry analysis and revised the paper, P.J.G. advised on the study, provided HIV-1 virus, and revised the paper, K.P. and F.S. performed experiments, S.G. advised on the study, generated and validated humanized mice for this study, L.G. generated and validated humanized mice for the study. All the authors have approved the final version of this paper.

## Competing interests
The authors declare no competing interests.
