## [Peer Review File · Communications Biology]

Reviewers' comments:

Reviewer #1 (Remarks to the Author):

The manuscript "Impaired extinction of cocaine seeking..." by Buck et al, has several major flaws and lacks innovation.

First, in terms of innovation, HIV and cocaine interactions have been extensively reported in the literature (Paris et al., 2014; Zhu et al, 2022; McLaurin et al., 2022). Thus, the manuscript lacks conceptual innovation.

Second, a flaw of the studies is illustrated in figure 1b. Approximately 10 of the HIV infected animals overlap with sham control levels of HIV copy number. Thus, a significant number of HIV-1 infected animals have no infection, and this is not accounted for in presenting the results.

Third, the manuscript conclusions are based on a single time point (day 3) in CPP extinction testing. The CPP behavioral test is generally regarded as difficult to interpret, as many factors can affect the testing outcome (i.e., motivation, learning, activity). In fact, one interpretation is that the HIV-infected animals actually remember the task better than the sham animals!

Fourth, the sample size in the behavioral studies is small (ns 4-5) relative to the number of animals initially infected.

Finally, the manuscript is poorly prepared: the sex of the animals is not reported, the photomicrographs are not publication quality, and the results of the cytokine/chemokine assays is over a page of F-values.

Reviewer #2 (Remarks to the Author):

This is an interesting report utilizing a very translationally-relevant humanized mice with active HIV infections to gain an understanding of the relationship between cocaine exposure and HIV on central and peripheral immune factors and reward-related behavior. Previous similar research has mainly examined the effect of specific HIV-related proteins rather than an actual infection. Surprisingly, HIV infection did not significantly alter (or augment) acquisition of cocaine CPP, but it did impair the extinction of that preference. The authors also looked at yohimbine induced reinstatement and cocaine seeking under conflict but found that neither was impacted by HIV infection. At the level of the immune response, the authors looked directly at histological markers of GFAP and Iba1 in the brain as well as at peripheral markers of inflammation (both mouse and human-derived) from the blood plasma. Brain region specific effects of HIV infection and/or cocaine were found for the histopathological markers. Cytokine and chemokine arrays pointed to many HIV-induced changes, but there were largely non-overlapping discoveries between the mouse and human panels. Overall, these results are interesting and novel and provide a fairly comprehensive analysis. I don't have major concerns about the paper or its contents. I do have a few questions/comments/suggestions for improvement of an already strong paper.

1. Please indicate where the humanized mice are sourced from. (Methods lines 88-89).
2. The manuscript does not suggest or indicate any premature mortality in these mice associated with any of the manipulations including the HIV infection, but these animals can have the tendency to become quite sick. Indeed supplementary diet gel is mentioned to help maintain healthy body weight. Please clarify if it is indeed the case that 100% of the animals made it through the entire study or whether there was any attrition.
3. It is interesting that both central and peripheral immune markers are probed in this paper. Could GFAP and Iba1, which are examined immunohistochemically, also be examined as circulating markers so that there is a subset of the same markers looked at in both compartments?
4. It is unclear if the cytokine data in particular (also may be relevant to IHC) has been adjusted for multiple comparisons or by using something like the false discover rate to limit type 1 error.

5. What is the time point of the viral load data in Figure 1B? What did it look like across time given that samples were collected every other week?
6. In figure 8, panels F-I are not mentioned anywhere in the text. There are also some spacing optimization things that could be done with the figures for final presentation like removing 8E from the main figure.
7. Please indicate the breakdown of male and female mice across experiments and groups. I appreciate that it was indicated that no sex differences were observed and understand that the numbers were overall unequal, but it would still be helpful to know the breakdown.
8. For the less well-versed, a more detailed explanation of the difference between G-CSF and GM-CSF would be useful in the discussion section as well as elaboration on the GM-CSF and interferon gamma relationship.
9. High rates of stereotypy are mentioned briefly in the discussion. Was there any evidence that this could have impacted the CPP behavior? Was this observed more in one group over another- like associated with the HIV infection?

Reviewer #3 (Remarks to the Author):

This is a well-designed study that investigated the effects on HIV infection on cocaine related behavior. The manuscript is well written, easy to read and the findings are well presented.

I just have a few questions related to time of experiments. It is well known that changes in circadian rhythms have an effect on CPP in mice. Could you clarify the times in the light dark cycle as well as when CPP occurred. Was it the same time of day for each experimental paradigm.

I was also curious to know why authors settled on a ip dose of cocaine at 10mg/kg as well as the concentration of yohimbine (2mg/kg). Since adrenergic receptors are known to be expressed on immune cells, does this have any effect on cytokine production, it maynot?

We appreciate the feedback of the reviewers, which we have incorporated as described below. We hope that this improved manuscript is now suitable for publication. Changes to the text are indicated in blue.

Reviewer #1 (Remarks to the Author):

The manuscript "Impaired extinction of cocaine seeking..." by Buck et al, has several major flaws and lacks innovation.

1. First, in terms of innovation, HIV and cocaine interactions have been extensively reported in the literature (Paris et al., 2014; Zhu et al, 2022; McLaurin et al., 2022). Thus, the manuscript lacks conceptual innovation.

We appreciate the reviewer's point to highlight additional literature in this space, and indeed, we had cited two of these references (Paris, et al., 2014 and Zhu et al., 2022) already in the manuscript, and now have incorporated the McLaurin manuscript to the discussion. While we fully recognize that the current work is an addition to an already meaningful and substantial literature on the importance of intersecting effects of cocaine and HIV, we respectfully disagree on the value of this work to the field as this is the first to report behavioral and immune outcomes in this mouse model of HIV-1 infection. We discuss areas of overlap with our findings and the extant literature, which we believe points to areas of robust overlap to contextualize these findings with the expectations that common vs novel findings across multiple models of co-occurring HIV infection and cocaine exposure will inform these fields independently and at their intersection.

2. Second, a flaw of the studies is illustrated in figure 1b. Approximately 10 of the HIV infected animals overlap with sham control levels of HIV copy number. Thus, a significant number of HIV-1 infected animals have no infection, and this is not accounted for in presenting the results.

We appreciate the opportunity to provide additional clarity around these results in the revised figure presentation. We now include both time course data, as requested by Reviewer 2 (new Fig 1d), and the mean viral copy data (now Fig 1c), and indicate both the mean and 95% CI for the sham animals to demonstrate that we did not include any animals whose mean viral load data across testing overlapped with the numbers observed in our sham animals. We hope this clarification alleviates these concerns.

3. Third, the manuscript conclusions are based on a single time point (day 3) in CPP extinction testing. The CPP behavioral test is generally regarded as difficult to interpret, as many factors can affect the testing outcome (i.e., motivation, learning, activity). In fact, one interpretation is that the HIV-infected animals actually remember the task better than the sham animals!

We agree that the CPP task reflects multiple components of reward-related behavior and that extinction learning reflects a competition between a previous memory and the new extinction memory. We added a line to the discussion highlighting this potential interpretation. We further agree that subsequent work will benefit from self-

administration studies. We did not observe differences in locomotor behavior during the CPP test that would drive these alterations (new Figure 2d, 4d).

The following has been added to the discussion (lines 344-347):

Persistent responding in the CPP task may also reflect greater reward-context conditioning in the initial CPP training that was not reflected in the CPP scores, or alternatively, differences in cocaine-seeking motivation.

4. Fourth, the sample size in the behavioral studies is small (ns 4-5) relative to the number of animals initially infected.

We regret the lack of clarity around our experimental design and figure presentation. After infection, mice were assigned to one of two behavioral groups – either the CPP extinction and reinstatement test *or* the CPP under conflict test. The n's for these behavioral experiments are indicated in the figure legends.

5. Finally, the manuscript is poorly prepared: the sex of the animals is not reported, the photomicrographs are not publication quality, and the results of the cytokine/chemokine assays is over a page of F-values.

While sex was previously reported only in the methods and discussion, we now indicate sex in each of the individual data point figures through use of different symbols. We apologize that the results section is cumbersome to read, however, believe that transparent reporting of statistics is essential for evaluation by other experts in the field and for interpretation of the results. We have selected only a subset of results to report in the results section that we believe are essential to the manuscript, while others are in the tables. We have opted to maintain this within the results as we think that this improves interpretability of the figures.

Reviewer #2 (Remarks to the Author):

This is an interesting report utilizing a very translationally-relevant humanized mice with active HIV infections to gain an understanding of the relationship between cocaine exposure and HIV on central and peripheral immune factors and reward-related behavior. Previous similar research has mainly examined the effect of specific HIV-related proteins rather than an actual infection. Surprisingly, HIV infection did not significantly alter (or augment) acquisition of cocaine CPP, but it did impair the extinction of that preference. The authors also looked at yohimbine induced reinstatement and cocaine seeking under conflict but found that neither was impacted by HIV infection. At the level of the immune response, the authors looked directly at histological markers of GFAP and Iba1 in the brain as well as at peripheral markers of inflammation (both mouse and human-derived) from the blood plasma. Brain region specific effects of HIV infection and/or cocaine were found for the histopathological markers. Cytokine and chemokine arrays pointed to many HIV-induced changes, but there were largely non-overlapping discoveries between the mouse and human panels. Overall, these results are interesting and novel and provide a fairly comprehensive analysis. I don't have major

concerns about the paper or its contents. I do have a few questions/comments/suggestions for improvement of an already strong paper.

1. Please indicate where the humanized mice are sourced from.

The methods section now indicates that the humanized mice were sourced from the University of Nebraska Medical Center (lines 88-89).

2. The manuscript does not suggest or indicate any premature mortality in these mice associated with any of the manipulations including the HIV infection, but these animals can have the tendency to become quite sick. Indeed supplementary diet gel is mentioned to help maintain healthy body weight. Please clarify if it is indeed the case that 100% of the animals made it through the entire study or whether there was any attrition.

We regret the lack of clarity around this. A subset of animals were removed from the studies. Any mice that did not survive the entirety of relevant behavioral testing were removed from data analysis. Additionally, mice with a high degree of sickness or stereotypic behavior were also excluded from analysis. This is indicated in the Methods (lines 97-99).

3. It is interesting that both central and peripheral immune markers are probed in this paper. Could GFAP and Iba1, which are examined immunohistochemically, also be examined as circulating markers so that there is a subset of the same markers looked at in both compartments?

We agree that this is of interest, but do not currently have sufficient sample availability to generate these results and thus regrettably cannot include it in the revised manuscript.

4. It is unclear if the cytokine data in particular (also may be relevant to IHC) has been adjusted for multiple comparisons or by using something like the false discover rate to limit type 1 error.

We have now applied a false discovery rate correction to the cytokine array data. This has resulted in restructuring, in particular for results of the mouse cytokine array data where results did not survive FDR correction. Results are now moved to a supplement (Supplemental Table 1) for potential interested readers, but are not discussed in the main results. A majority of the human cytokine array markers remained significant following FDR correction, and this figure has been revised accordingly and split into two figures for ease of communication (now Figures 8 and 9)

5. What is the time point of the viral load data in Figure 1B? What did it look like across time given that samples were collected every other week?

We have now updated the original figure to reflect mean viral load data for all animals (now Fig 1c) and included time course data for HIV-1 inoculated mice (new Fig 1d).

6. In figure 8, panels F-I are not mentioned anywhere in the text. There are also some spacing

optimization things that could be done with the figures for final presentation like removing 8E from the main figure.

The figures presenting the cytokine array results have been substantially restructured following the FDR corrections.

7. Please indicate the breakdown of male and female mice across experiments and groups. I appreciate that it was indicated that no sex differences were observed and understand that the numbers were overall unequal, but it would still be helpful to know the breakdown.

We have updated individual data points to indicate sex in the majority of figures wherever possible (i.e., when individual data points are shown).

8. For the less well-versed, a more detailed explanation of the difference between G-CSF and GM-CSF would be useful in the discussion section as well as elaboration on the GM-CSF and interferon gamma relationship.

Because the results/discussion were restructured following application of the FDR correction, we have removed discussion of G-CSF from the discussion and have not expanded this section. We have added an expanded discussion on GM-CSF and interferon gamma and potential implications for individuals with a history of cocaine exposure (lines 447-456).

9. High rates of stereotypy are mentioned briefly in the discussion. Was there any evidence that this could have impacted the CPP behavior? Was this observed more in one group over another- like associated with the HIV infection?

Mice with abnormal stereotypic behavior were removed from the study prior to completion of experiments – now indicated in the methods-, and did not differ based on infection status.

Reviewer #3 (Remarks to the Author):

This is a well-designed study that investigated the effects on HIV infection on cocaine related behavior. The manuscript is well written, easy to read and the findings are well presented.

I just have a few questions related to time of experiments. It is well known that changes in circadian rhythms have an effect on CPP in mice. Could you clarify the times in the light dark cycle as well as when CPP occurred. Was it the same time of day for each experimental paradigm.

All experiments were performed in the light cycle at approximately the same time of day. The following statement has been added to the methods (lines 116-117):

All behavioral training and testing took place during the light cycle, between zeitgeber times 6 and 9, Monday through Friday.

I was also curious to know why authors settled on a ip dose of cocaine at 10mg/kg as well as the concentration of yohimbine (2mg/kg). Since adrenergic receptors are known to be expressed on immune cells, does this have any effect on cytokine production, it maynot?

The dose of cocaine (10 mg/kg) was selected as it consistently yields CPP in mice. Similarly, the yohimbine dose was selected based on previous publication. References and a statement indicating this have been added to the text in the Methods section.

The point on the adrenergic effect on cytokine production is a very interesting one, and something that we hope to consider further in the future. We have included a statement on this in the discussion as an important consideration/caveat (lines 441-446).

REVIEWERS' COMMENTS:

Reviewer #2 (Remarks to the Author):

The authors have addressed all of my prior concerns/comments.

Reviewer #3 (Remarks to the Author):

Authors have answered my queries appropriately.

Manuscript is much clearer written in response to all reviewers critiques.